# A Collagen-Mimetic Organic-Inorganic Hydrogel for Cartilage Engineering

**DOI:** 10.3390/gels7020073

**Published:** 2021-06-15

**Authors:** Laurine Valot, Marie Maumus, Luc Brunel, Jean Martinez, Muriel Amblard, Danièle Noël, Ahmad Mehdi, Gilles Subra

**Affiliations:** 1IBMM, University Montpellier, CNRS, ENSCM, 34095 Montpellier, France; laurine-valot@laposte.net (L.V.); luc.brunel@umontpellier.fr (L.B.); jean.martinez@umontpellier.fr (J.M.); muriel.amblard@umontpellier.fr (M.A.); 2ICGM, University Montpellier, CNRS, ENSCM, 34095 Montpellier, France; 3IRMB, University Montpellier, INSERM, CHU Montpellier, 34090 Montpellier, France; marie.maumus@inserm.fr; 4Bauerfeind France, IRMB, 34090 Montpellier, France

**Keywords:** hydrogel, sol-gel, hybrid material, collagen-mimetic peptide, cartilage tissue engineering, mesenchymal stromal cells

## Abstract

Promising strategies for cartilage regeneration rely on the encapsulation of mesenchymal stromal cells (MSCs) in a hydrogel followed by an injection into the injured joint. Preclinical and clinical data using MSCs embedded in a collagen gel have demonstrated improvements in patients with focal lesions and osteoarthritis. However, an improvement is often observed in the short or medium term due to the loss of the chondrocyte capacity to produce the correct extracellular matrix and to respond to mechanical stimulation. Developing novel biomimetic materials with better chondroconductive and mechanical properties is still a challenge for cartilage engineering. Herein, we have designed a biomimetic chemical hydrogel based on silylated collagen-mimetic synthetic peptides having the ability to encapsulate MSCs using a biorthogonal sol-gel cross-linking reaction. By tuning the hydrogel composition using both mono- and bi-functional peptides, we succeeded in improving its mechanical properties, yielding a more elastic scaffold and achieving the survival of embedded MSCs for 21 days as well as the up-regulation of chondrocyte markers. This biomimetic long-standing hybrid hydrogel is of interest as a synthetic and modular scaffold for cartilage tissue engineering.

## 1. Introduction

The treatment of articular cartilage injuries remains challenging due to the inherent nature of the tissue and its complex structural organization. Many research efforts have focused on cartilage tissue engineering during the last decades, mainly focusing on the use of mesenchymal stem or stromal cells (MSCs) to repair the degraded cartilage. Nevertheless, the regeneration of mature articular cartilage instead of a repair with a tissue of an inferior quality made of fibrocartilage or hypertrophic cartilage still represents the real issue for cartilage regenerative medicine. Promising results have been obtained when cells were incorporated into a scaffold [1,2,3,4]. Compared with the injection of cells alone, scaffolds improved the cell viability and differentiation and could be used to fill cartilage focal defects [5]. Different types (natural, synthetic) and forms (membrane, sponge) of biomaterials have been tested for cartilage repair but hydrogel scaffolds are particularly suitable for an application with cell encapsulation, substituting the natural extracellular matrix (ECM) during the development of the new cartilage. Developing such a scaffold is not an easy task as the material should ideally reach the mechanical performance required for a resistance to the compressive and shear stress of the joint while presenting the biocompatibility [6], degradability, stability and porosity required to reach the chondrosupportive and/or chondroinductive properties [7].

Many strategies have been reported relying on physical and chemical hydrogels made from collagen [8,9,10,11], gelatin [12], hyaluronic acid [13,14,15,16,17,18], a decellularized ECM [19,20,21,22,23], chitosan [24,25] or hydroxyl-propylmethyl cellulose (HPMC) [26,27]. The use of a synthetic polymer such as polyethylene glycol (PEG) [28,29,30,31,32] is attractive from a batch-to-batch reproducibility point of view and also allows a straightforward control of the network properties (i.e., size of the mesh, cross-linking density), along with an ease of functionalization. Synthetic polymers, however, do not recapitulate the features of the main components of an ECM for cartilage engineering.

Peptides comprising (Pro-Hyp-Gly) repeated sequences have been widely investigated as synthetic collagen surrogates and can turn into hydrogels if they reach a suitable size and by adding a hydrogelator part to their sequence (i.e., Phe-Phe parts, palmitoyl or cysteines) [33,34,35,36,37,38]. Unfortunately, these are physical hydrogels [39] that lack the stability required for a long-lasting cell encapsulation and the mechanical properties for resisting the high mechanical constraints applied to the joints. Chemical hydrogels including collagen-mimetic peptides and encapsulating cells were prepared using the photo-polymerization of PEG acrylate derivatives [40]. However, they were not the main component of the network and the photo-polymerization could induce the production of toxic free oxygen radicals. Moreover, this type of polymerization could induce an inhomogeneity of the peptide distribution inside the 3D network and thus a low control of the mechanical and biological properties all along the scaffold.

We first demonstrated that inorganic sol-gel polymerization could be used as a biorthogonal reaction proceeding at a physiological pH [27,40,41]. A structure driven assembly had already been exploited to prepare the hybrid material [42]. More specifically, we also showed that the sequence of hybrid silylated peptides could drive the assembly of a 3D network that was further covalently fixed by the establishment of siloxane bonds [43]. Combining these interesting features (i.e., self-assembly, siloxane covalent bond formation as a chemoselective and biorthogonal reaction), we designed and synthesized bis-silylated collagen-like undecapeptides (Ac-Lys-[Pro-Hyp-Gly]_3_-Lys-NH_2_) to prepare a chemical hydrogel by a sol-gel process in the presence of murine MSCs [40]. Beyond this proof-of-concept, several limitations remained. First, at a physiological pH, the optimal sol-gel reticulation required non-biocompatible amounts of fluoride as a catalyst. Lowering the quantity of NaF to improve the viability (0.01% weight) led to a slow gelation (> 24 h). Secondly, the quantity of hybrid peptide required to form the hydrogel was quite high (10% weight) compared with the collagen-based hydrogels, which contain 0.3% weight collagen. The length of peptides could explain this; they were not long enough to provoke a triple helix assembly, which could have helped to generate an additional network organization and non-covalent reticulation. Thirdly, we observed that embedded cells were unable to spread or proliferate and the viability decreased after one week of culture. We hypothesized that the modulation of the mechanical properties and the sagging of the network could improve the long-term viability and facilitate the cell differentiation.

To tackle these issues, we investigated longer peptide building blocks to favor the triple helix formation. The 6-pattern collagen-like peptides [Ac-Lys-(Pro-Hyp-Gly)_6_-Lys-NH_2_], showed a triple helix self-assembly and yielded hydrogels at a lower concentration than the 3-pattern peptides (6 wt% vs. 10 wt%) [44]. We also applied a glycine/NaF co-catalysis protocol to reduce the gelation time without any toxicity. Finally, the co-addition of mono-silylated collagen-mimetic peptides allowed a fine-tuning of the mechanical properties. Human MSCs (hMSCs) encapsulated within these novel hydrogels stayed alive for 21 days and expressed chondrocyte markers when induced to differentiate in a chondrogenic medium.

## 2. Results and Discussion

### 2.1. Synthesis and Characterization of the Hybrid Peptides

To demonstrate that hybrid hydrogels prepared from collagen-like peptides could be an alternative to natural biopolymers, it was of importance to study the scale-up of our synthetic strategy. Our process was adapted to synthesize up to 600 mg of collagen-like peptides in an SPPS reactor, thanks to a convergent method. After a preparative RP-HPLC purification, the peptides were obtained with a 48 and 49% yield, respectively (purity > 99%). Indeed, a first block corresponding with the protected (permanent and temporary) tripeptide collagen motif Fmoc-Pro-Hyp(tBu)-Gly-OH (**PM**) was synthesized at a 15 g scale (Figure 1). The protected peptide was then used as a building block along with Fmoc-Lys(Boc)OH to synthesize collagen-like peptides **6M2K** and **6M1K** (Figure 1). Of note, the coupling of PM proceeded without any risk of epimerization as it presented a glycine as the C-terminal residue.

Collagen-like peptides were then functionalized with one or two triethoxysilyl moieties by reacting isocyanides with the primary amines of the lysine side chain(s), leading to the formation of one or two urea functions (Figure 2). This quantitative reaction proceeded in anhydrous conditions under argon with an excess of a base (DIEA). No addition of ICPTES on the hydroxyproline side chain was observed in the NMR spectra (Appendix A). Hybrid peptides **6M-2Si** and **6M-1Si** were recovered by precipitation as white powders and used without any further purification.

The secondary and tertiary structures of the peptides and their hybrid analogues (200 µM in water, pH 7.4 at 20 °C) were investigated by a circular dichroism. During the minutes of analysis and without the presence of any catalyst, no condensation reaction occurred. As expected, the CD spectrum of the four compounds showed a poly-proline II (PPII) secondary structure with a positive maximum at 225 nm and a negative maximum at 195 nm. However, this signature reflected only the PPII secondary structure and did not tell us about the collagen triple helix supramolecular assembly (Figure 3b). To decipher that feature, the Rpn (ratio of positive on negative value) corresponding with the ratio of the CD signals at 225 nm over 220 nm, was calculated (Figure 3a). The presence of a triple helix has already been admitted for Rpn values above 0.11 [34,45,46]. We have previously demonstrated that peptides containing three motives were too short to form a triple helix and showed an Rpn value in the range of 0.66 and 0.082 (naked or silylated respectively) [40]. The Rpn of **6M2K** was low (0.068) probably because of the two cationic residues flanking the repeated sequence, which might disrupt the triple helix assembly. Once silylated, the Rpn of **6M-2Si** increased to 0.099 indicating a higher supramolecular assembly content. The Rpn values of the mono-functionalized **6M1K** and **6M-1Si** compounds were slightly above 0.11 and were consistent with the presence of triple helixes. It is worth noting that higher Rpn values could be obtained by enlarging the peptide sequences (i.e., 7 Pro-Hyp-Gly motives or greater) as reported in the literature [45] but the solubility of such silylated hybrid peptides considerably dropped and did not allow the preparation of the hydrogels.

### 2.2. Hydrogel Formation, Gelation Time and the Addition of the Mono-Silylated Collagen-Like Peptide

The gelation of the hybrid collagen-like peptide solution was obtained by a sol-gel process by the concomitant hydrolysis of ethoxysilyl moieties into hydroxysilyl moieties and condensation into siloxane bonds, creating reticulation nodes [48]. In numerous biomaterial applications, sol-gel is either acid- or base-catalyzed and does not allow the entrapment of cells before a careful neutralization. By contrast, we developed here a biocompatible nucleophile catalytic system consisting of a low concentration of NaF (0.1 mg/mL) combined with glycine (10 mg/mL). We have already demonstrated this catalytic effect on PEG hydrogels [48] and HMPC alike [27]. We applied these experimental conditions to obtain the hybrid collagen peptide hydrogels.

We first studied the required concentration of **6M-2Si** to prepare the hydrogels at 37 °C at pH 7.4 in a DPBS or DMEM culture medium. Above 7 wt%, **6M-2Si** did not reach complete dissolution. However, at 4 wt% and below, the solution did not make a gel even after one week. At 6 wt%, **6M-2Si** yielded white opaque hydrogels in about 7 h (Table 1) so we chose this concentration for further assays. Of note, at this concentration the solution prepared from the mono-silylated **6M-1Si** hybrid peptide never reached gelation. This demonstrated the importance of chemical cross-linking in the establishment of the 3D network.

However, to favor cell survival and colonization of the network, the control of the hydrogel structure and the number of chemical reticulation nodes are of importance. For that purpose, we investigated the replacement of the increasing amounts of the bis-silylated peptide by the same amount of the mono-silylated one, which formed a triple helix assembly easier than the bi-functional counterpart (Figure 3). We hypothesized that this swap could facilitate cell spreading in the matrix. The 95/5, 90/10 and 85/15 (**6M-2Si**/**6M-1Si**) molar ratios were used, keeping the total amount of the hybrid peptide at 6 wt%. Gelation did not occur with the 85/15 ratio even after 24 h. By contrast and interestingly, hydrogels obtained with the 95/5 and 90/10 ratios turned into gels within 7 h yielding white, opaque and more fragile hydrogels compared with those obtained with the bis-silylated peptide only when manipulated.

### 2.3. Structural and Mechanical Characterization of the Hydrogels

The 6 wt% **6M-2Si** and **6M-2Si/6M-1Si** (9/1 molar ratio) hydrogels were then compared with the commercially available 0.3 wt% collagen type I hydrogel for the 3D cell culture. First, cryo-SEM analyses were performed (Figure 4). Interestingly, the synthetic network presented many similarities with the collagen I hydrogel network in particular, having macropores ranging from 2 to 5 µm and long interconnected fibers. In contrast, it is worth noting that hydrogels obtained from the silylated PEG and under the same experimental conditions were less fibrous with a low porosity. The average pore size of the synthetic networks was higher (~7 µm diameter) than the one found in the collagen network (~5 µm diameter). The same observation was made for the distance between the fibers. It is also worth noting that the addition of the mono-silylated peptide had no visual impact on the 3D network organization.

The rheological evaluation was carried out by indentation measurements. As expected, the addition of a 10% molar mono-silylated peptide improved the elasticity of the hydrogel, considering that the maximal stress before the rupture increased from 109 to 119 kPa (Figure 5a) and the maximal deformation increased from 48 to 59% of the thickness (Figure 5b). In comparison, the collagen I hydrogel was softer with a 14 kPa maximal stress and 68% maximal strain. This difference could be explained by a higher cross-linking of the chemical hydrogel yielding tougher materials. These higher values are of particular interest for cartilage engineering applications. From these results, the Young’s modulus was calculated, providing valuable information on the resistance of this hybrid biomimetic material (Figure 5c). The addition of the mono-silylated peptide led to a decrease in the Young’s modulus (from 103 to 79 kPa). However, these values were two to three times higher than those of the collagen I hydrogel (25 kPa). Even when these values were far from the ones of the native cartilage (1000–5000 kPa) [49], they were still higher than for the natural type I collagen hydrogel and were convenient for the cell encapsulation, allowing the cell proliferation and differentiation inside the hydrogel. The network mesh size (ξ), which monitors the physico-chemical properties of the hydrogel (rigidity, water displacement, etc.), could also be calculated from these values [50], giving information on the distance between the molecules inside the network. The addition of the mono-silylated peptide resulted in a small mesh size increase from 4.9 to 5.4 nm. This difference could be due to the lower number of reticulation nodes (Figure 5d). In comparison, the collagen I hydrogel formed by the self-assembly of fibrils presented a mesh value of 7.9 nm.

Swelling studies (Figure 6) were carried out to determine the amount of water that could be re-absorbed by freeze-dried hydrogels, which behaved like hybrid foams. Three foams obtained from the 6 wt% **6M-2Si** and **6M-2Si/6M-1Si** (90/10 and 95/15 molar ratio) were studied. Of note, the collagen I hydrogel proved to be impossible to compare with the synthetic hydrogels in this swelling assay. Indeed, once freeze-dried, the collagen I hydrogel lost its structure and dissolved totally in water in contrast to the synthetic hybrid chemically cross-linked biomaterials, which were able to swell and keep their shape.

All of the three hybrid freeze-dried hydrogels behaved similarly, re-uptaking 76–78% water of their initial water content. No difference was observed after 8 and 24 h of swelling, meaning that equilibrium was already reached within 8 h. We also calculated the network mesh size (ξ) from the swelling assays [51], which was about 2.5 nm in the same order of magnitude than ξ calculated by rheology (approximations were made for the calculations, see Appendix A).

Summing up, the addition of the mono-silylated collagen-like peptide **6M-1Si** had almost no impact on the 3D network, either on the mesh size or on the swelling properties. However, it changed the rheological properties, decreasing the Young’s modulus and yielding to a more elastic hydrogel.

### 2.4. Biological Evaluation of the Hydrogels for Cell Encapsulation

To evaluate the biocompatibility of the **6M-2Si** and **6M-2Si/6M-1Si** hydrogels, hMSCs (1.10^6^ cells/mL) were encapsulated inside the hydrogel during the gelation process. The cell viability was evaluated one day after encapsulation by confocal microscopy and a LIVE/DEAD cell viability assay. As shown in Figure 7, the majority of hMSCs in the **6M-2Si** and **6M-2Si/6M-1Si** hydrogels were alive (green color) and few cells were dead (red color). No difference was observed between the **6M-2Si** and **6M-2Si/6M-1Si** hydrogels. The cell viability was also obtained in the 0.3 wt% collagen type I hydrogel commonly used for the 3D cell culture. It is worth noting that in the **6M-2Si** and **6M-2Si/6M-1Si** hydrogels, hMSCs exhibited a round shape compared with the collagen type I hydrogel in which they exhibited a fibroblastic phenotype. These observations suggested that hMSCs were entrapped inside the network of the **6M-2Si** and **6M-2Si/6M-1Si** hydrogels and were not adherent compared with their behavior in the collagen type I hydrogel. Due to the similarity between the two hydrogels and the slightly higher pore size of the **6M-2Si/6M-1Si** hydrogels, further experiments were performed with the **6M-2Si/6M-1Si** hydrogel.

To confirm these observations, cryo-SEM images of **6M-2Si** + **6M-1Si** (6 wt% 90/10) and 0.3 wt% collagen I hydrogels with encapsulated hMSCs were taken after one day of culture and formaldehyde fixation (Figure 8). In the two samples, hMSCs were entrapped inside the hydrogel and kept a round shape. As expected, the collagen I hydrogel appeared more fibrillary than the synthetic hybrid collagen-like hydrogels. Both hydrogels surrounded the cells, creating a 3D network around them.

The long-term viability was then evaluated in the **6M-2Si/6M-1Si** hydrogels for 21 days of culture. As shown in Figure 9a, the cell number decreased after 21 days of culture compared with day 1. A quantification of the DNA content confirmed a decreased cell number at day 21 (Figure 9b). This decrease in cell number after 21 days of culture was not reported in a study using PEG-L-PA co-polymer-based hydrogels where cells were cultured in an FCS-containing proliferative medium [52].

Finally, the chondroconductive properties of the **6M-2Si/6M-1Si** hydrogels were evaluated by inducing the differentiation of hMSCs by culture in a chondrogenic medium containing the chondroinductive factor TGF-β3. After 21 days of differentiation, the control hMSCs cultured in micropellet conditions were able to differentiate toward chondrocytes as they exhibited a significant increase of chondrogenic markers *SOX9*, *ACAN*, *COL2A1* and *COL10A1* compared with undifferentiated hMSCs at day 0 After 21 days of culture in **6M-2Si/6M-1Si** hydrogels in a chondrogenic medium with TGF-β3, hMSCs expressed lower levels of *SOX9* and *ACAN* than undifferentiated hMSCs. However, they expressed higher levels of the chondrogenic markers *SOX9*, *ACAN*, *COL2A1* and *COL10A1* compared with the cells in a chondrogenic medium without TGFβ3 (negative control) (Figure 9c).

## 3. Conclusions

Collagen hydrogels have been widely used for tissue engineering approaches for diverse applications. However, they present several limitations that may impair their use for biomedical purposes. First, they come from animal sources and may suffer from low batch-to-batch reproducibility and safety issues. Second, they lack proper mechanical properties enabling sustained cell differentiation and ECM production. Adding cross-linking agents could overcome this limitation at least partially but inherent toxicity is detrimental to cell viability.

As far as cartilage engineering is concerned, the encapsulation of MSCs requires a hydrogel, which has to be stiffer than collagen hydrogels to attain a matrix elasticity consistent with the MSC commitment towards chondrocyte differentiation. Moreover, cartilage ECM components are produced by hMSC-derived chondrocytes after two weeks, at least in vitro, indicating that the 3D matrix surrounding the cells should remain for a time sufficient to be replaced by the neo synthesized matrix. In these conditions, biomimetic synthetic hydrogels could be valuable. We have demonstrated that a combination of short peptides bearing one or two cross-linkable alkoxysilane moieties could yield interesting hydrogels whose mechanical properties are situated between the native cartilage and collagen I hydrogels. Sharing a comparable microstructure with collagen I hydrogels, such hybrid biomimetic hydrogels are able to maintain the long-term survival of encapsulated MSCs and to provide a chondroconductive environment that allows the up-regulation of chondrocyte specific genes when cultured in a chondroinductive medium. Although improvements are still needed, we bring the proof-of-concept that biomimetic peptide-based hydrogels may allow the commitment of hMSCs towards the chondrocyte lineage. Interestingly, they could also be turned into foams by freeze-drying, which opens up potential applications as implantable scaffolds that could be colonized by endogenous cells.

## 4. Materials and Method

### 4.1. Materials

All reagents and solvents were purchased from Alfa Aesar (Kandel, Germany), Acros (Illkirch, France), Iris Biotech (Marktredwitz, Germany), Sigma–Aldrich and Merck (Lyon, France) and were used without further purification.

### 4.2. Peptide Synthesis

Peptides were prepared by a solid phase peptide synthesis (SPPS) following the Fmoc/tBu strategy using DMF as a solvent. Detailed SPPS protocols are available in the Appendix A.

Briefly, the protected tripeptide building block Fmoc-Pro-Hyp(tBu)-Gly-OH (peptide motif, noted **PM**) was synthetized on a 2.2-Cl-Chlorotrityl PS resin. The **PM** was purified by a preparative HPLC before being used as building block for the synthesis of Ac-Lys-[Pro-Hyp-Gly]_6_-Lys-NH_2_ (**6M2K**) and Ac-[Pro-Hyp-Gly]_6_-Lys-NH_2_ (**6M1K**) on a hydrophilic PEG-containing resin bearing a Fmoc-Rink amide linker (amphisphere 40 RAM resin). Both peptides were purified by a preparative HPLC before silylation and were analyzed by LC-MS (see Appendix A).

### 4.3. Peptide Silylation on the Lysine Side Chains

Bis- and mono-silylated hybrid peptides **6M-2Si** and **6M-1Si** were obtained from **6M2K** and **6M1K**, respectively, by reacting with 3-isocyanatopropyltriethoxysilane (ICPTES) following a previously described procedure [40].

Briefly, the peptide was dissolved in an anhydrous DMF (1 g/7 mL) under argon. Diisopropylethylamine (DIEA) and ICPTES were added in excess and the mixture was stirred at room temperature for 1 h 30 min (see Appendix A for the detailed procedure). The reaction mixture was precipitated in diethyl ether. The resulting solid was washed 3 times with the precipitation solvent and dried under a vacuum. The hybrid peptide was then stored at 4 °C under argon.

Hybrid peptides were analyzed by LC-MS using water/acetonitrile 0.1% TFA and by ^1^H, ^13^C and ^29^Si NMR performed in a deuterated DMSO (see Appendix A).

### 4.4. Circular Dichroism Analyses

The secondary and tertiary structures of peptides and hybrid peptides were assessed by circular dichroism (CD) in water pH 7.4 at 200 µM after 1 h of solubilization at 20 °C. The CD analyses were recorded on a Jasco J-815 spectropolarimeter (Lisses, France) with the wavelength range set to 190–280 nm and a scanning speed of 100 nm/min in a 0.1 cm path length quartz cell. Data were acquired over three scans and the background signal (water) was subtracted from the obtained values.

### 4.5. Preparation of the Hybrid Hydrogels

The hybrid peptide **6M-2Si** was poured in Dulbecco’s Phosphate Buffered Saline (DPBS) pH 7.4 (6 wt%; a volume of 1 mL was commonly used) containing NaF and glycine (respectively, 0.1 g/L and 10 g/L). After the complete dissolution, the solutions were filtered through a 0.2 µm filter for sterilization. The gelation times were determined at 37 °C using the tilting method; i.e., the gelation was observed when the sample could not flow any more upon the inversion of the vial.

### 4.6. Cryo-SEM Images

Hydrogel embedding cells were prepared with the desired compositions in Eppendorf tubes. They were allowed to make gel for one day at 37 °C before fixation in 4% formaldehyde. Control gels without cells and non-fixed gels were made to check that formaldehyde fixation had no impact on the hydrogel microstructure. The samples were kept at 4 °C before being frozen in a nitrogen slush (−210 °C). The metallization was performed with gold-palladium. For a better observation of the cells, the metallization was performed before freeze-drying to protect the cell’s structure. A cryo-SEM analysis was performed on a Jeol 6700F equipped with a cryo-transfer EDS (PLACAMAT platform, Bordeaux University, CNRS, France).

### 4.7. Indentation Measurements

The macro-indentation experiments were performed on hydrogels (1.20 mL) poured into a 12-well plate after six days of aging at 37 °C in a humid atmosphere. The experiment was done in duplicate. The indentation test was performed with a duralumin cylinder connected at one end to the stress-controlled rheometer (HR-2 rheometer from TA Instruments) and presented a circular flat surface on the other end (diameter 10 mm). The speed of descent was set to 0.1 mm/s until a 200 µm gap (≈ 92% of strain) was reached. The normal force sensor of the rheometer was used as a force gauge to determine the stress-strain relation during the sample indentation. The maximum normal stress (σmax) and compressive strain (εmax) could be read when the hydrogel cracked under compression, leading to a fall in the σ = f(ε) curve due to the rupture (Appendix A). In the first moments of the deformation (under 5% of deformation), the linear low-strain regime provided a way to define an apparent elastic modulus (E * = δσ/δε) and, after calculation, the Young’s modulus E and then the storage modulus by calculation from E. Finally, the mesh size could be calculated (see Appendix A).

### 4.8. Swelling Studies

The hydrogels (700 µL) were prepared in 3 mL syringes and aged for five days at 37 °C. They were cut into three pieces (cylinder of around 3 mm height and 4 mm radius) as triplicates and weighed (m_hydrogel_) before freeze-drying and weighed again (m_dried_). They were then immersed in water (1 mL) at room temperature. Prior to being weighed after 1 h and 24 h of swelling (m_wet_), the rehydrated hydrogels were quickly blotted with filter paper to remove the excess surface water. The water uptake was calculated as 100 xm_wet_/m_hydrogel_. The mesh size of the hybrid foams and hydrogels were then calculated from this ratio following equations described in the Appendix A (see Appendix A and Appendix A).

## Figures and Tables

**Figure 1 gels-07-00073-f001:**
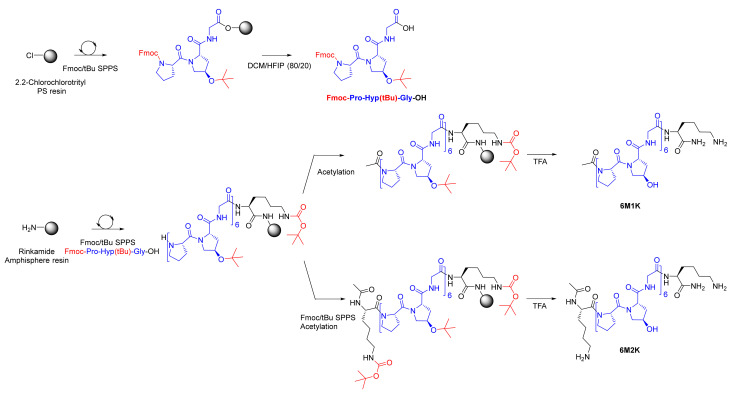
Synthesis of a PM block (in blue) and **6M1K**/**6M2K** collagen-like peptides.

**Figure 2 gels-07-00073-f002:**
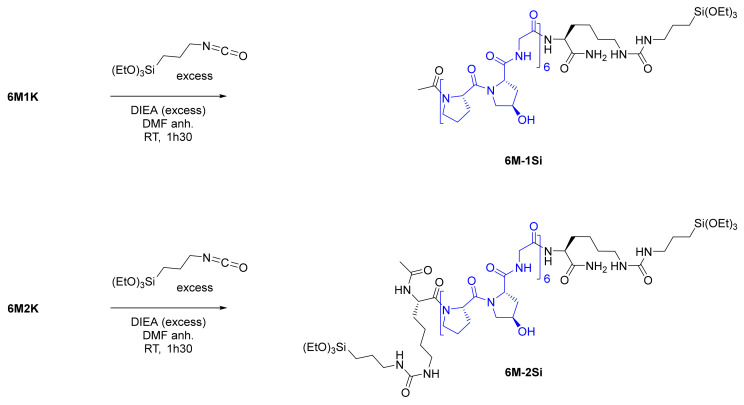
Silylation of **6M1K**/**6M2K** yielding the **6M-1Si**/**6M-2Si** hybrid collagen-like peptides.

**Figure 3 gels-07-00073-f003:**
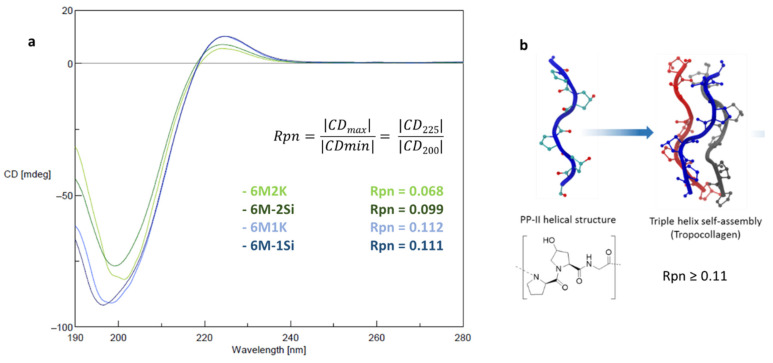
(**a**) CD spectrum of collagen-like peptides and their hybrid relatives with the calculated Rpn value and (**b**) schematic representation of PPII and triple helix formation (adapted from [47]).

**Figure 4 gels-07-00073-f004:**
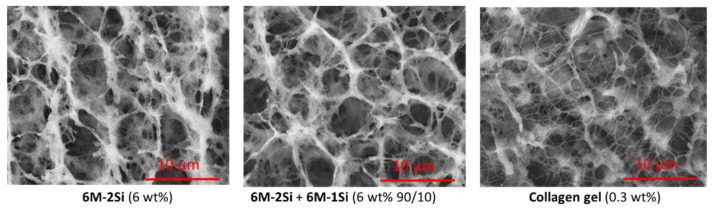
Cryo-SEM images of hydrogels obtained at 37 °C in DPBS at pH 7.4 for 24 h.

**Figure 5 gels-07-00073-f005:**
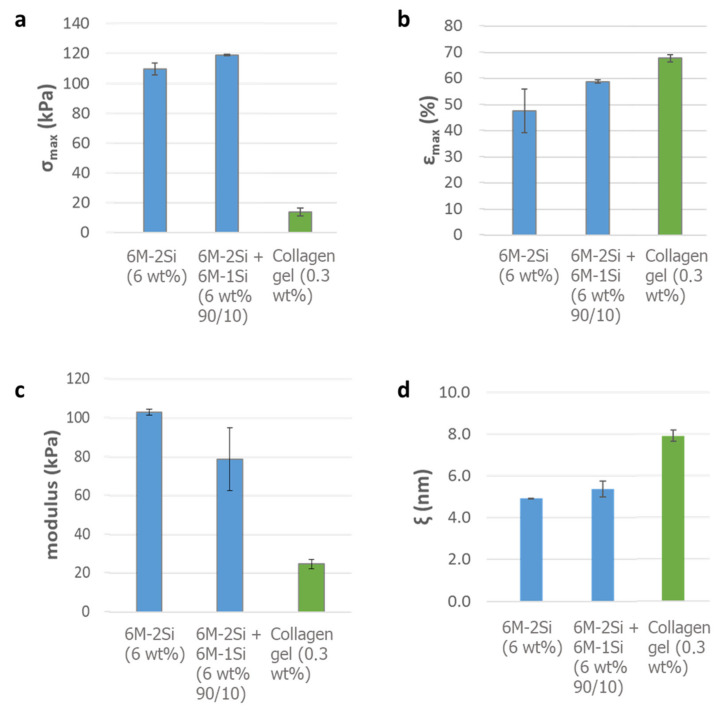
Rheological evaluation of the collagen type I hydrogel and hybrid hydrogels obtained with 0.1 mg/mL NaF and 10 mg/mL glycine at 37 °C in DPBS pH 7.4 and aged for six days. (**a**) Maximal stress before rupture; (**b**) maximal strain before rupture; (**c**) Young’s modulus; (**d**) mesh size calculated by rheology.

**Figure 6 gels-07-00073-f006:**
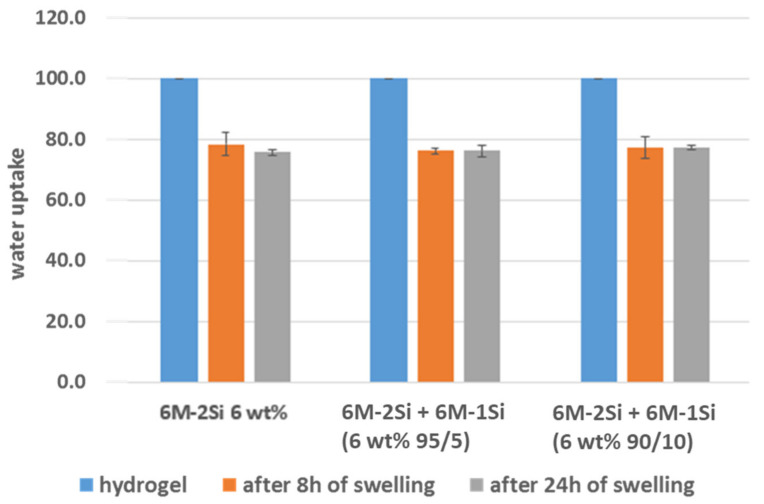
Water re-uptake of freeze-dried hydrogels made with 0.1 mg/mL NaF and 10 mg/mL glycine after 8 and 24 h at 37 °C in DPBS pH 7.4 aged for six days.

**Figure 7 gels-07-00073-f007:**
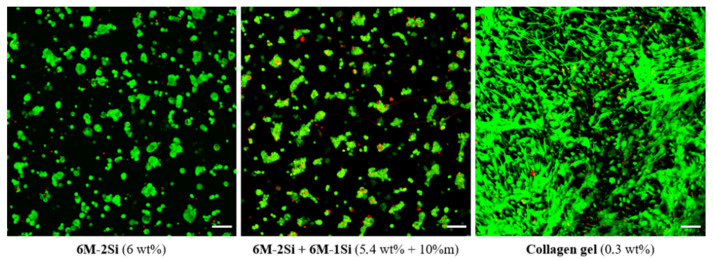
Cell viability of hMSCs measured by LIVE/DEAD staining in different hydrogels after one day of encapsulation. Images are the maximum intensity z-projection. Scale bar is 100 μm.

**Figure 8 gels-07-00073-f008:**
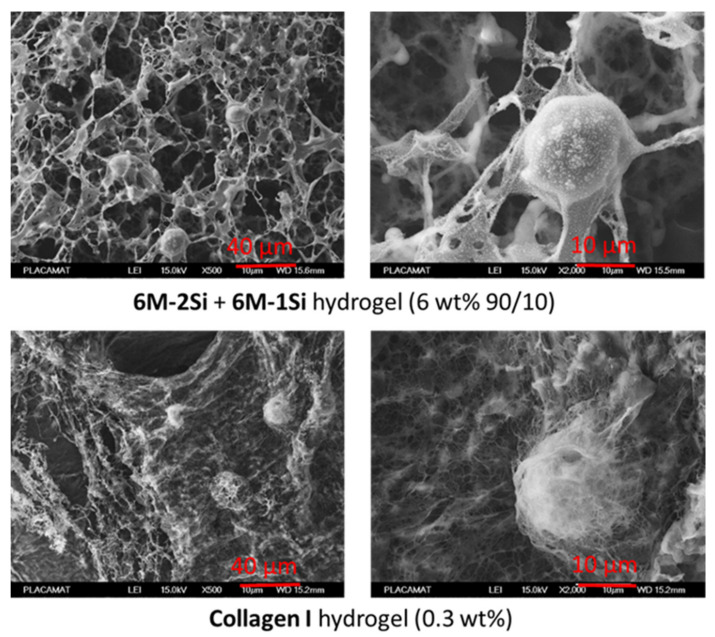
Cryo-SEM images of hydrogels made with 0.1 mg/mL NaF and 10 mg/mL glycine at 37 °C in DPBS pH 7.4 with 10^6^ hMSCs/mL one day after encapsulation.

**Figure 9 gels-07-00073-f009:**
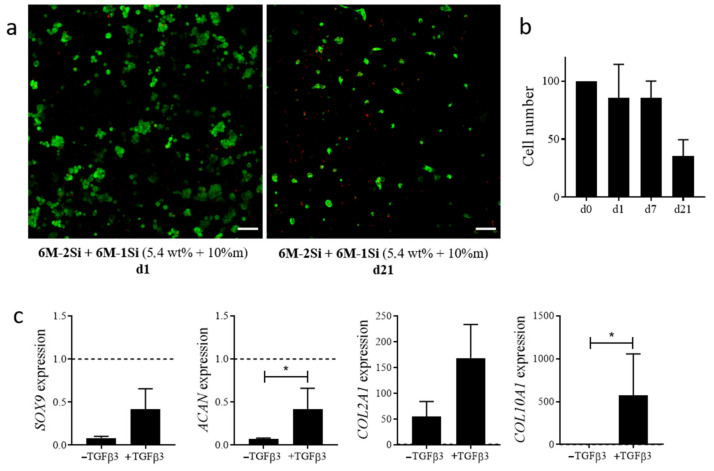
(**a**) Cell viability of hMSCs measured by the LIVE/DEAD assay in **6M-2Si**/**6M-1Si** hydrogels after 1 and 21 days of encapsulation. Images are the maximum intensity z-projection. Scale bar is 100 μm. (**b**) Number of hMSCs measured by DNA quantification in a **6M-2Si**/**6M-1Si** hydrogel at day 0, 1, 7 and 21 days after encapsulation. Data are expressed as the mean ± SEM of four independent experiments and normalized to 100% at day 0. (**c**) Chondrocyte gene expression in hMSCs encapsulated in **6M-2Si**/**6M-1Si** hydrogels after 21 days of differentiation. Data are expressed as the mean ± SEM of three independent experiments and expressed in a fold change compared with undifferentiated hMSCs at day 0 (dotted line). Statistical analyses were performed using GraphPad prism 9 software with the non-parametric Mann-Whitney test to compare the two conditions. Values are statistically different when *p* < 0.05 (* *p* < 0.05).

**Table 1 gels-07-00073-t001:** Gelation time and the aspect of hydrogels with 0.1 mg/mL NaF and 10 mg/mL glycine at 37 °C in DPBS pH 7.4.

Hybrid Peptide	Concentration	Gelation Time	Hydrogel Aspect
**6M-2Si**	7 wt%	- ^a^	Above solubility
**6M-2Si**	6 wt%	6 h 30 min–7 h 30 min	White opaque
**6M-2Si**	5 wt%	> 8 h	Very weak, white opaque
**6M-2Si**	4 wt%	- ^b^	-
**6M-2Si + 6M-1Si**	6 wt% (95/5)	6 h 30 min–7 h	White opaque
**6M-2Si + 6M-1Si**	6 wt% (90/10)	6 h 30 min–7 h	White opaque
**6M-2Si + 6M-1Si**	6 wt% (85/15)	- ^b^	-

^a^ Hybrid building block was not fully soluble. ^b^ No gelation observed after 24 h.

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
