# Peer review of "A Collagen-Mimetic Organic-Inorganic Hydrogel for Cartilage Engineering"

_gels, 2021, doi:10.3390/gels7020073_

Round 1
Reviewer 1 Report
see attached.

Author Response
See cover letter

Reviewer 2 Report
In this contribution by Valot and Maumus, and co-workers, the authors developed a novel collagen-mimetic organic-inorganic hydrogel for cartilage regeneration. The results are interesting and potentially attractive to the readership of Gels. However, some corrections are needed in order to increase the overall quality of the paper before publication.
- Many errors occur possibly due to refs in the section of ‘Results and discussion’.
- The quality of figure 3,5, and 6 should be improved (including the resolution of the images).
- One scheme to show how the hydrogel formed by self-assembly and biorthogonal sol-gel cross-linking should be added.
- No figure 9 is found in the ms.
- Cell number decreased after 21 days of culture compared to day 1 significantly. The authors should compare the results with recent studies (doi.org/10.1016/j.actbio.2020.04.027; org/10.1021/acs.biomac.0c00623) where not significant decreasing of cells was found.
Author Response
See cover letter please

Round 2
Reviewer 2 Report
I recommend it for publication in the current form.